# Pattern of New Gene Origination in a Special Fish Lineage, the Flatfishes

**DOI:** 10.3390/genes12111819

**Published:** 2021-11-19

**Authors:** Haorong Li, Chunyan Chen, Zhongkai Wang, Kun Wang, Yongxin Li, Wen Wang

**Affiliations:** School of Ecology and Environment, Northwestern Polytechnical University, Xi’an 710072, China; lainnorton@outlook.com (H.L.); chenchunyan@nwpu.edu.cn (C.C.); 15834186753@163.com (Z.W.); wangkun@nwpu.edu.cn (K.W.); yxli28science@sina.com (Y.L.)

**Keywords:** flatfishes, phylogenomic context, origin, new genes

## Abstract

Origination of new genes are of inherent interest of evolutionary geneticists for decades, but few studies have addressed the general pattern in a fish lineage. Using our recent released whole genome data of flatfishes, which evolved one of the most specialized body plans in vertebrates, we identified 1541 (6.9% of the starry flounder genes) flatfish-lineage-specific genes. The origination pattern of these flatfish new genes is largely similar to those observed in other vertebrates, as shown by the proportion of DNA-mediated duplication (1317; 85.5%), RNA-mediated duplication (retrogenes; 96; 6.2%), and de novo–origination (128; 8.3%). The emergence rate of species-specific genes is 32.1 per Mya and the whole average level rate for the flatfish-lineage-specific genes is 20.9 per Mya. A large proportion (31.4%) of these new genes have been subjected to selection, in contrast to the 4.0% in primates, while the old genes remain quite similar (66.4% vs. 65.0%). In addition, most of these new genes (70.8%) are found to be expressed, indicating their functionality. This study not only presents one example of systematic new gene identification in a teleost taxon based on comprehensive phylogenomic data, but also shows that new genes may play roles in body planning.

## 1. Introduction

New genes may play significant roles in evolution of organisms [1,2]. The term “new gene” refers to a novel genetic unit that originated in a specific lineage [2]. As they only exist in certain species or evolutionary lineages, new genes are also called species or lineage-specific genes [3]. The main origin mechanisms of new genes include DNA-mediated duplication, RNA-mediated duplication and de novo origination, which can be identified from their sequence features [4]. With large number of genomes being sequenced, comparative genomic methods can be used to identify new genes based on the species phylogeny [5,6]. The synteny-based pipeline (SBP) method [7] is suited for recently duplicated genes’ identification, whereas the protein-family-based methods are useful for ancient new genes [8,9]. The establishment of these method provides us an opportunity to understand the pattern of origin of new genes in a lineage.

So far, many studies on new genes have focused on the model organisms including primates, rodents and *Drosophila* [1,10,11,12,13]. However, there are few researches on new genes in non-model vertebrates, and no study using phylogenomic data has been conducted to investigate new genes in a lineage of teleosts [14]. In addition, teleosts usually have an additional genome-wide duplication event, compared to other vertebrates, which may result in more duplicate genes and some of them may have functions in these fishes [15,16]. Among teleosts, flatfishes have evolved a very unique body plan, especially for its asymmetric body structure has been of great interest [17,18]. Recently, a relatively complete set of genome data of flatfishes and their outgroup species were published by us [19,20], providing a valuable opportunity to comprehensively identify new genes and investigate their possible roles in the evolution of flatfishes.

In this study, we defined the gene age using *Platichthys stellatus* (starry flounder) as the focal species representing the real flatfishes (Pleuronectoidei), which has the highest quality of genome assembly in flatfishes. Traditionally flatfishes include Pleuronectoidei and Psettodoidei species, but here we only consider the Pleuronectoidei, because Pleuronectoidei and Psettodoidei do not form a monophyletic group and Psettodoidei has only one species, *Psettodeserumei* [19]. We reconstructed the phylogenetic tree and estimated the divergence time in order to identify new genes by the SBP method. The results show that the flatfish-specific new genes have similar evolutionary pattern to those identified in primates, and perhaps due to longer divergence time (73.7 Mya) higher proportion of new genes of flatfishes were under functional constraint. The divergence time between the reference species (*P. stellatus*) and its closest species (*Paralichthys olivaceus*) is around 41.8 Mya. In addition, we observed some new genes were asymmetrically expressed in the two body sides during the metamorphosis of the starry flounder, indicating their possible roles in the development of special body plan of flatfishes.

## 2. Materials and Methods

### 2.1. Data Acquisition

Four published high-quality Pleuronectoidei genome assemblies, including starry flounder (*P. stellatus*, GCA_016801935.1), Japanese flounder (*P. olivaceus*, GCF_001970005.1), turbot (*Scophthalmus maximus*) and tongue sole (*Cynoglossus semilaevis*, GCF_000523025.1) [19], were collected to represent the species in Pleuronectoidei. The genome assembly of turbot was downloaded from the website (https://denovo.cnag.cat/turbot) [21]. The genome assembly of two Carangarians, which are the closest species of Pleuronectoidei, including sailfish (*Istiophorus platypterus*, GCA_016859345.1) and swordfish (*Xiphias gladius*, GCF_016859285.1) [20], and two close Perciformes species, yellow perch (*Perca flavescens*, GCF_004354835.1) [22] and Arkansas darter (*Etheostoma cragini*, GCF_013103735.1) [23], were retrieved from NCBI and used as outgroup species.

### 2.2. Phylogenetic Reconstruction

After collecting the protein-coding genes in the eight species, the longest transcript for each gene was retained. All-vs-all BLAST alignment was used to find the relationship of reciprocal best BLAST hit among genes, which were used to construct the phylogenetic tree by RAxML (v8.2.10) [24] with the PROTGAMMAAUTO model. The fourfold degenerate synonymous sites (4dTVs) were extracted and prepared from the genes to estimate the divergence time of species using the MCMCtree model in PAML (v4.4) [25] with the calibration of several fossil records downloaded from the TimeTree website (http://www.timetree.org accessed on 25 February 2021).

### 2.3. Identification of New Genes

Synteny-based pipeline (SBP) [7] was used to identify the new genes of Pleuronectoidei. Considering the quality of genome assembly, we selected the chromosome-level starry flounder genome generated by the third-generation sequencing [18] as the reference in the subsequent analysis. The Lastz software (v1.04.00) [26] was used to align each collected genome to the reference genome. Parameters used in Lastz were set differently according to the divergence time of the query species with starry flounder, which refers to the different species alignment parameters with human in UCSC (https://genome.ucsc.edu/index.html, accessed on 25 February 2021). Then the reciprocal alignment results were generated following the steps in Zhang et al. [27]. The branch of new gene origination was inferred by the distribution of their position and existence in whole genome reciprocal alignment results. The age of each gene can be marked using branches 0–5. The genes that exist in all species on the phylogenetic tree were defined as the oldest genes and the branch is 0, while the branch 5 indicates genes only existing in the starry flounder. If there are multiple exons or transcripts of a gene, the age of the oldest exon or transcript was used to represent the gene [7]. In addition, genes with more than 70% overlapping sequences with repetitive sequence regions or are not located on the chromosomes were excluded. To detect possible horizontal gene transfer, especially for those non-duplicated de novo genes, the NR (Non-Redundant Protein Sequence) database of NCBI was used to align flatfish new genes with the genes of microorganisms. None of the new genes is homologous to that of microorganisms, and for all of them homologous sequences can be found in the outgroup teleosts, suggesting that there was no horizontal gene transfer event happened in the origination of flatfish new genes.

### 2.4. Inference of the Origin Mechanism

The origin mechanisms of new genes were mainly classified into DNA-mediated duplicate genes, RNA-mediated duplicate genes (retrogenes), and de novo genes. The method in the previous studies [27,28] was used to analyze the origin mechanism of new genes. To further check the absence of homologous genes of de novo genes in outgroup species, NR database and Pfam databases were further checked.

### 2.5. Selection Analysis of New Genes

Selection signal of duplicate genes was calculated by comparing their closest paralogous genes in the starry flounder. The relationship between duplicate genes and their paralogous copies was obtained by BLAST (v2.2.26). The MUSCLE (v3.8.3) [29] software was used to align their protein sequences, and then pal2nal [30] was used to convert the results to codon level alignment. The PAML (v4.9e) [25] software was used to calculate the *Ka*/*Ks* value. The likelihood ratio (LTR) test was executed to calculate *p*-value with the assumption of *Ka*/*Ks* = 0.5 [31]. According to Betrán et al. [31], a paralogous *Ka*/*Ks* ratio significantly (by likelihood ratio test; LRT) lowers than 0.5 is considered as evidence for evolutionary constraint on both copies. The genes with *Ka* > 0.5, *Ks* > 5 or the *Ks* value beyond the 1.5-flod interquartile range in the *Ks* distribution were filtered. The candidate paralogous genes with outlier *Ks* values may not represent real paralogous genes. Duplicate genes with *Ka*/*Ks* < 0.5 and *p*-value < 0.05 were considered to be under negative selection.

We used the codeml of PAML4.9 [25] package to calculate the *Ka*/*Ks* values and performed the likelihood ratio test (LRT) with the assumption of *Ka*/*Ks* = 1 to test whether a de novo gene is functionally constrained by natural selection, we calculated the *Ka*/*Ks* ratio and *p*-value between a de novo gene and its homologous genes in Japanese flounder genome, if there exists. If *Ka*/*Ks* < 1 and *p*-value < 0.05, respectively, they were considered to be under selective pressure. To do this, we first used BLAST (v2.2.26) to align de novo genes to Japanese flounder genome and extended 10 Kb of upstream and downstream of homologous regions. Next, we used protein sequence of de novo gene as template to predict open reading frame (ORF) of homologous sequences by Exonerate (v2.2.0) [32]. The longest ORF without stop codon or frame shifts was selected.

### 2.6. Gene Expression in Tissues of Starry Flounder

We downloaded the published multi-tissue RNA-sequencing (RNA-seq) data of starry flounder from NCBI, which have two replicate data for male and female brain, gonad, liver and muscle (PRJNA556158), respectively. We also downloaded RNA-seq data for the eyes and body tissues at the post-metamorphosis stage of starry flounder larvae (PRJNA592732), including the left-eye type’s tissues around the eye, the left-eye type’s around both eyes, the right-eye type’s tissues around the eye and the right-eye type’s around both eyes. Unfortunately, these RNA-seq data of starry flounder were not generated from left and right side of an individual, rather both side tissues from an individual were mixed, although left type and right type individuals were sampled separately. Therefore, theses data cannot be used to look at the asymmetrical expression of new genes, and can be only used to check if a new gene was expressed or not. The adaptors and low-quality bases were removed with Trim-galore (v0.5.0) [33]. Hisat2 (v2.2.1) [34] was used to align the RNA-seq reads to the starry flounder genome with default parameters. Then the Stringtie (v2.1.4) [35] was used to quantify the transcripts to obtain the fragments per kilobase million (FPKM) values and confirm the expression of new genes.

### 2.7. Asymmetrical Expression Analysis

The published multi-tissue RNA-seq data of Japanese flounder was downloaded from NCBI with accession number of PRJNA632737. There are 72 RNA-seq libraries which were produced by the method of orthogonal design, including four developmental stages around metamorphosis (pre-metamorphic larva; pro-metamorphic larva; metamorphic climax larva; post-metamorphic larva), three tissues (eye; muscle; skin), and two sides (left side; right side). Each sample was constructed and sequenced for three biological replicates. Hisat2 (v2.2.1) [34] was used to align the RNA-seq reads to the Japanese flounder genome with parameters—very-sensitive. The Stringtie (v2.1.4) [35] software was used to quantify the transcripts to obtain FPKM values for the 200 Pleuronectoidei-specific new genes (branch 2-4) which are also present in the Japanese flounder. The raw count values generated by Stringtie were used for the differential expression analysis of the left and right side data of each stages by the R package Deseq2 (v1.28.1) [36]. It is noteworthy that the genome of the Japanese flounder was assembled using short next generation sequencing reads [37] and, thus, was not good enough to be used as the reference genome when we were identifying new genes.

## 3. Results

### 3.1. New Genes Emerged in Pleuronectoidei

We re-constructed the phylogenetic tree and estimated the divergence time in order to identify new genes by the SBP method [7,27,28]. The divergence time between the reference species (*P*. *stellatus*) and its closest species is around 41.8 million year ago (Mya), and in the flatfishes with outgroups is 73.7 Mya. Then, there are 1541 Pleuronectoidei-specific new genes (assigned on branch 2–5; Appendix A) were identified, which account for 6.9% of the genes that were located on chromosomes of the starry flounder (Figure 1). The species-specific genes emergence rate is 32.1 Mya and the whole average level rate for the Pleuronectoidei lineage genes is 20.9 per Mya. There are 1317 (85.5%) DNA-mediated duplicates, 96 (6.2%) RNA-mediated duplicates (retrogenes) and 128 (8.3%) de novo genes. We did not detect any new genes originated from horizontal gene transfer. The proportions of different categories of new genes are consistent with previous results in mammals [7,28,38]. Indicating the pattern of new gene origination in Pleuronectoidei is largely similar to other animal taxa [39] and that the ancient fish-specific genome duplication event may not retain many duplicates in the modern teleosts [39,40].

### 3.2. Most New Genes Are Expressed and Some under Natural Selection

About 74.0% (1046 out of all the 1413 duplicate genes, including both the DNA and RNA mediated duplications) of new duplicate genes are found to be expressed in at least one tissue (FPKM > 0.5; Figure 2B). After filter genes with outlier *Ks*, 308 of the 1413 duplicate genes can be conducted using paralog *Ka*/*Ks* analysis. Among the 308 genes, 128 (31.4%) genes were shown to be under negative selection (*Ka*/*Ks* < 0.5; *p*-value < 0.05) (Figure 2A), of which 101 genes were also expressed. In a previous study on primates [7], the proportion of old genes under negative selection was significantly higher than the new duplicate genes (65.0% in old genes and 4.0% in new genes). In our study, the proportion of old genes (branches 0–1) under negative selection was 66.4% (1299 genes under negative selection/1964 old genes), but the proportion of new duplicate genes under selection in our study is much higher, which may be explained by the deeper divergence of Pleuronectoidei (~73 Mya [15]) compared to the primates (~43 Mya [7]), or the new duplicate genes in Pleuronectoidei might have been under higher selection pressure.

There are 128 de novo genes in Pleuronectoidei, of which 45 genes were found to be expressed in the analysis of the transcriptome (FPKM > 0.5), accounting for 35.2% of the de novo genes in Pleuronectoidei. We counted the expression of all de novo genes in various tissues (Figure 2C), and found that the number of genes expressed in all tissues is 14, the number of genes expressed in only a single tissue is 5, and the others expressed in two or more tissues. The expression of de novo genes varied in different tissues, with the highest number of 61 genes expressed in male gonad and lowest number of 19 genes expressed in female gonad. The result of more de novo genes expressed in male gonads is consistent with the observation in other animals by the previous study [7], but the specific functions of these de novo genes need further investigation. Among the 128 de novo genes, through *Ka*/*Ks* analysis with the orthologous genes in Japanese flounder, we found only two genes have been under significant negative selection, including gene evm.model.Hic_chr_10.1034 (*Ka*/*Ks* = 0.16; *p*-value = 1.4 × 10^−2^) and gene evm.model.Hic_chr_16.310 (*Ka*/*Ks* = 0.13; *p*-value =2.8 × 10^−5^). Among them, the gene evm.model.Hic_chr_10.1034 was expressed in all tissues (Appendix A), which might have evolved into a housekeeping gene in Pleuronectoidei.

By analyzing RNA-seq data in nine tissues of starry flounder, we found that gene transcription profiles changed across different gene age groups. The median expression of young genes (Pleuronectoidei-specific genes; branch 2–5) is close to 0 FPKM, while the median of nearly half of the old genes (branch 0–1) is higher than 1.0 FPKM (Figure 2C). In the expression profiles of various tissues, the young genes (branch 2–5) are expressed with a median less than 1.0 FPKM in 3 tissues, while the old gene (branch 0–1) are expressed with a median higher than 2.0 FPKM in 8 tissues. This age-related expression trend was also observed in the results of previous studies in primate- or rodent-specific genes [4,7].

### 3.3. Asymmetric Expression of Some New Genes May Play Roles in the Formation of the Unique Body Plan of Flatfishes

To investigate the possible roles of new genes on the asymmetric body plan of the Pleuronectoidei, unfortunately, we lacked two sides’ transcriptome data of starry flounder from same individuals (see Methods). We thus tentatively used the transcriptome data from Japanese flounder for differential expression analysis for the 200 Pleuronectoidei-specific new genes which are also present in Japanese flounder (branch 2–4; Figure 3A). By comparing the expression level of left and right side in each stage, we found that the stage and tissue containing the most number of differentially expressed genes (DEGs) was the pro-metamorphic larva of muscle (Appendix A). In other tissues, only one or zero new genes were significantly differentially expressed. In these DEGs at pro-metamorphic larva of muscle, nine new genes were significantly differentially expressed (*p*-value < 0.05, Appendix A), eight genes are DNA-mediated new duplicate genes, and one gene is de novo gene (evm.model.Hic_chr_6.523) which has no functional annotation information yet.

Among the new genes with significantly higher expression on the right side, one gene is *Hipk1* (Figure 3B), which is related to the regulation of eyeball size, lens formation and retinal morphogenesis [41], and we found 14 genes that belonged to *Hipk* gene family in the Pleuronectoidei-specific new genes. The *Hipk* gene family were also found to have flatfish-specific expansion in previous study [37]. Besides, two new genes associated with cell proliferation were also highly expressed on the right side, including *Nlrc3-like* [42] (Figure 3C) and *Trim25* [43] (Figure 3D). Asymmetric expression of these three new genes may have potential functions in the formation of the asymmetry of the left and right sides of the Japanese flounder.
Figure 3New genes expressed in Japanese flounder. (**A**) Heatmap of expression of 200 Pleuronectoidei-specific new genes which are present in Japanese flounder in left and right side muscle at different development stages, Pre, pre-metamorphic larva; Pro, pro-metamorphic larva; Clim, metamorphic climax larva; Post, post-metamorphic larva. Genes were clustered by hclust [44]. (**B**–**D**) Expression level and *p*-value of genes *Hipk1*, *Nlrc3-like* and *Trim25* at the pro-metamorphic larva of muscle.
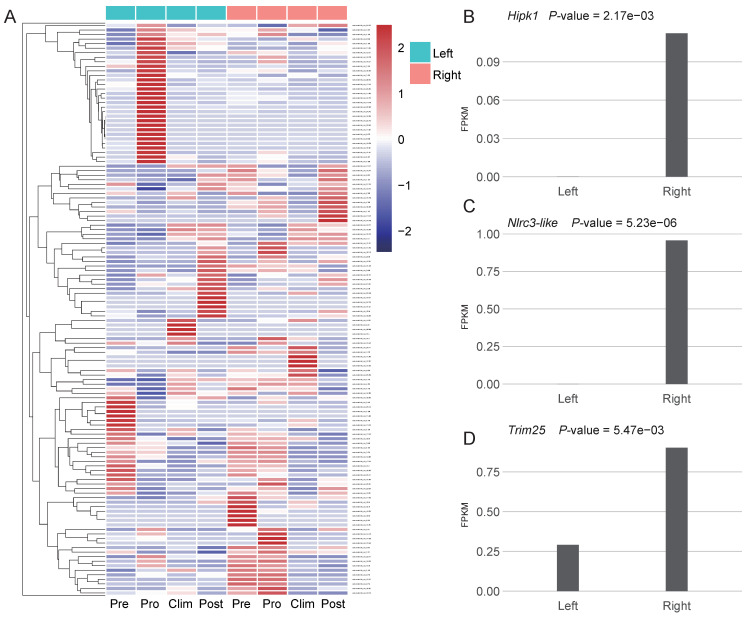



## 4. Discussion

Our results show that origin of new genes in flatfishes shares similar pattern and mechanism as observed in other animals including mammals [7,28,45]. They originated mainly by DNA-mediated duplicates (85.5%), with some were from RNA-mediated duplicates (retrogenes) (6.2%) or de novo genes (8.3%), indicating that different animal taxa share similar patterns of new gene origination. In fishes, it is somewhat unexpected that the proportions of these categories of new genes are similar to previous results observed in mammals [7,28,38], suggesting that the ancient teleost-specific genome duplication event did not retain many duplicates in the modern teleosts. 

In this study we found a high proportion (1341/1541 = 87%) of species-specific genes in the focal reference starry flounder, which corresponds to a new gene emergence rate of 32.1 genes per Mya, in contrast to the whole average level rate for the Pleuronectoidei lineage (20.9 genes per Mya), indicating rapid birth and death rate of genes in the flatfish lineage. A previous study has found a higher zebra fish-specific gene duplication rate (9.0 genes per Mya) than Tetraodon (2.6 genes per Mya) [39]. The authors explained this phenomenon as distant divergence between the two fishes, and especially that zebra fish might have experienced more recent duplications [39]. In the case of the starry flounder, we also suspect that the long divergence time (41.8 Mya) between it and its closest species (the Japanese flounder) may partially lead to identification of more species-specific genes. However, it is possible that the higher genome assembly quality of the starry flounder than other flatfishes may artificially resulted in identification of some starry flounder-specific genes. Future studies with denser representative species and higher genome assemblies will clarify this problem.

It is interesting that the proportions of new genes under selection between Pleuronectoidei species and primates are highly different (31.4% vs. 4%) while the old genes remain quite similar (66.4% vs. 65%) [7]. This may have resulted from three possible reasons. First, flatfishes might have received stronger selection due to adaptation to the seafloor habitats. Second, the divergence time between Pleuronectoidei species and outgroup (73.7 Mya) is much earlier than that between primates and other mammals (43.0 Mya) [46], which thus allows us to see more selection-retained functional new genes in flatfishes. The third reason may be because that the focal reference species (the starry flounder) in this study diverged from its closest species (the Japanese flounder) 41.8 Mya, which is much earlier than that between human and chimpanzee (6.0 Mya), many species-specific new genes in the starry founder have evolved substantial functions and thus are under selection. In other words, many new genes identified in the primate study [7] are very young human-specific genes that have not evolved substantial functions and may disappear in the future. We think the third reason is the most important factor that has led to the observation of much higher proportion of new genes under selection in flatfishes.

In our transcriptome analysis, our results show that the new genes in fishes also tend to express in the male gonad, further supporting the conjecture that new genes tend to evolve in male reproduction-related functions [2,7,47]. The male gonad expressed de novo gene evm.model.Hic_chr_10.1034 is under negative selection, and can be partially (Identities = 24/95, 25%) aligned to deoxynucleotidyltransferase terminal-interacting protein 2 (*Dnttip2*). This gene may have relationship with chromatin remodeling and gene transcription [48]. This ubiquitously expressed de novo new gene deserves more functional studies in the future to demonstrate how a de novo gene can become a seemly essential gene.

Although only limited transcriptome data available from left vs. right sides’ tissues during the metamorphosis stages of a founder, we were still able to observe some new genes show asymmetric expression between the left and right tissues during metamorphic stages, which suggests they might be involved in the asymmetric body plan formation in flatfishes. In the future, collecting transcriptome data of left and right side of the metamorphosis stage from the reference species, the starry flounder, and especially more experimental validation work on these new genes, will provide more information to test the roles of these new genes in the asymmetric body plan of flatfishes.

## Figures and Tables

**Figure 1 genes-12-01819-f001:**
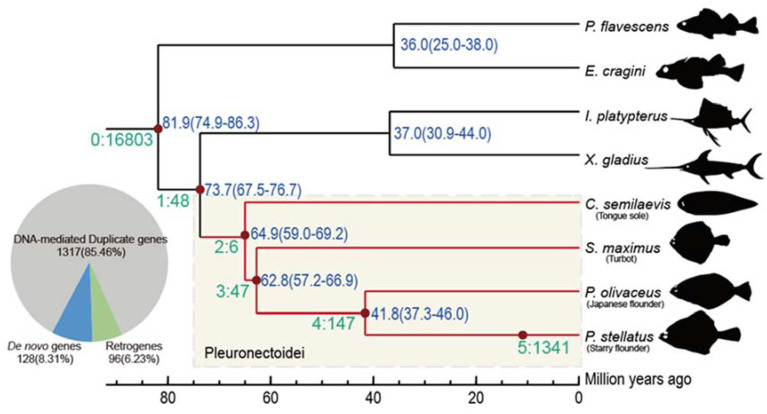
Branch view of the distribution of new genes. In SBP, a total of six distinct age groups (branch 0–5) were specified based on the phylogenetic context. Genes postdating *P. stellatus*-*C. semilaevis* split (branch 2–5) are referred as Pleuronectoidei-specific genes, marked with light yellow block. The number and proportion for each origin mechanism of the 1541 Pleuronectoidei-specific genes are plotted in a pie chart.

**Figure 2 genes-12-01819-f002:**
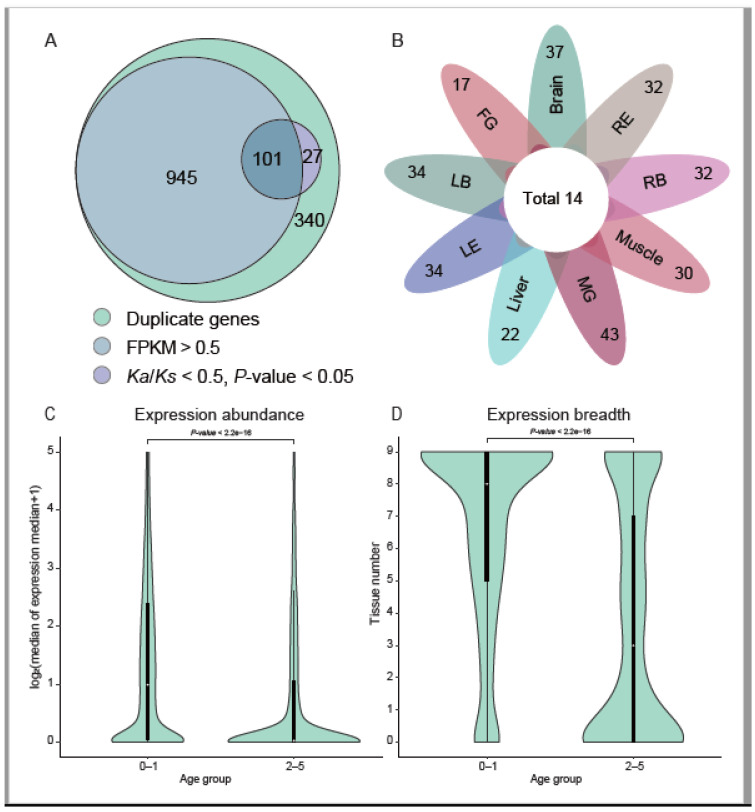
Expression of new genes. (**A**) Expression evidence of new genes. New genes with evidence of expression mean expressed in at least one tissue (FPKM > 0.5). Genes with significant paralog *Ka*/*Ks* < 0.5 are 128 (101 + 27). (**B**) Expression of de novo genes in different tissues. Each petal represents the number of genes expressed in the tissue (left-eye type’s tissues around eyes (LB), left-eye type’s two eyes (LE), right-eye type’s tissues around eyes (RB), right-eye type’s two eyes (RE), female gonad (FG), male gonad (MG). Fourteen de novo genes are expressed in all the studied tissues. (**C**,**D**) show violin plots of expression abundance and breadth of Pleuronectoidei-specific genes across indicated age groups (log2-based median of the median expression in 9 starry flounder tissues and the numbers of tissues where genes are expressed (FPKM > 0.5). In each case, the violin curve indicates the probability density of the data, the black bar in the center shows the interquartile range and the white dot shows the median. The genes were divided into two age groups. Results of Wilcox tests of the significance of differences between the age groups are also presented.

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
