# Peer review of "Pattern of New Gene Origination in a Special Fish Lineage, the Flatfishes"

_genes, 2021, doi:10.3390/genes12111819_

Round 1

Reviewer 1 Report

This study implements a classical pipeline to identify lineage-specific genes in flatfishes by using newly released genome sequences. These lineage-specific genes are also called new genes or young genes. They represent species/lineage specific genetic innovation and play a critical function in species/lineage specific traits or developmental effects such as body planning in flatfishes. New genes identification requires high-quality genome sequences of a group of phylogenetically related species. Therefore, new genes identification is often confined to limited lineages, especially this teleost is rarely reported. I appreciate the author increasing our knowledge of new genes origination patterns of this intriguing lineage. It's impressive that the high proportion and rapid origination of species-specific new genes in starry flounder indicate new genes play a critical role in species specialization. Even more interesting, the proportion of new duplicate genes under selection in flatfish is much higher than the Primates underlying deep divergence of Pleuronectoidei. These new discoveries undoubtedly expand our knowledge of the new gene origination of this special lineage. The asymmetric expressed new genes identified by this study lay a good foundation for underlying the molecular mechanism of body plan development in flatfish.

Since some of the writing seems succinct, many important discoveries and analyses are not adequately represented in this study. I encourage the author to revise their manuscript based on the following comments.

Minor revision:

  1. I would like to rephrase the wording “origin” to “origination” since “origin” refers to the beginning of something while “origination” basically means a process, e.g., “Pattern of new genes origination in a special fish lineage - the flatfishes”.
  2. The author may need to integrate more essential knowledge in the Abstract section based on the discovery in this study, e.g., total new genes proportion, the percentage, and gene number of the three typical mechanism patterns, the divergent time between the focus species and its closest species, the divergent time of P. stellatus-C. semilaevis (new genes definition time by this study), expression proportion of total new genes.
  3. Line 45-50, the author should increase 2-3 sentences to briefly describe the main work and discovery of this study including the phylogenetic tree and updated divergence time.
  4. Line 153, the author mentioned that they did not detect any new genes originating from horizontal gene transfer, I didn’t find a detailed description of how the author conducted this analysis. I would suggest the author describe it in the Materials and Methods section.
  5. The author should emphasize the high species-specific new genes proportion in this study, which is 1341/1541=87%. Additionally, the new gene emergence rate (per MY) comparison between the whole average level and species-specific level should also be included in both Abstract and Results.
  6. Line 169-174, It’s intriguing that the proportion of new genes under selection between Pleuronectoidei and Primates is highly different (4% vs 31.4%) while the old genes remain quite similar. The author should include this revelation in the Abstract. Also, if possible, the author should discuss this important discovery in terms of the high rate of species-specific new genes emergence in flatfish.
  7. The author should also list the Fold change values of the 9 DEGs between the left and right sides in Table S1.
  8. Figure 3A seems not to be cited.
  9. Figure S1 title is missing.
  10. The author should include a detailed list of those 1541 new genes in flatfish.

Reviewer 2 Report

The authors report results from their study on the evolution of new genes in Pleuronectoidei. Using a relatively complete set of genome assemblies representing Pleuronectoidei and four outgroup assemblies. The authors identified new genes through whole genome alignment and inferred putative origination mechanisms using previously reported methodologies. Tests for selection were conducted on the identified new genes followed by gene expression analyses in various tissues of Starry and Japanese Flounder.

The authors clearly present their rationale, findings, and future plans to incorporate more transcriptomic data to evaluate the role of new genes in body plan evolution. The discussion of methodologies is mostly clear. There is some vagueness regarding the selection analysis. Slightly modifying sections 2.5 and 3.2 to be more clear would be a great benefit to this already solid article. 

Lines 103-105 would benefit from justifying why these ratios were used (briefly explain what they are); there is a small typo on line 104. Lines 109-114 would benefit from rewording. As they are now they're very concise with some awkward phrases, ''the selection test of  . . . was analyzed with the orthologous genes . . .''  Lines 109-111 could be more clear as the phrasing doesn't explain what analysis particularly was conducted.  Section 2.5 would benefit from explaining what you are using LRT for. 

Line 166 needs to be reworded as the meaning is not immediately clear. Lines 207-209 need to be rewritten as ''...median of nearly half'' and ''...while more than half of the genes . . . were transcribed in at least 8 tissues'' are rather clunky to read/parse through.

A supplementary PCA plot (or similar) showing how left versus right side expressed samples cluster would strengthen these results. 

Author Response

This manuscript is a resubmission of an earlier submission. The following is a list of the peer review reports and author responses from that submission.

Round 1

Reviewer 1 Report

Li et al. investigated omics data from flatfish species, focusing on their evolutionary mechanism of asymmetric body plans. Using a synteny-based pipeline, they identified >1,500 lineage-specific genes that are largely under negative selection. With transcriptome data from the left and the right side tissues of Japanese flounder, the authors suspected that some of the lineage-specific genes are related to the promotion of asymmetric body plans of the flatfishes.

                 Asymmetric body plans are commonly observed among organisms. Among them, flatfishes are one of the extreme examples. Therefore, the research subject of the manuscript is very important and of broad interest. However, I am puzzled by the way the authors studied flatfishes in the manuscript. For example, there are many previous studies and reviews regarding asymmetry in vertebrates (e.g. Hamada et al. 2002, Nat Rev Genet 3:103-113), but discussion and acknowledgement on them are almost entirely missing in the manuscript. In Introduction, they rather narrowly focused on the evolutionary roles of new genes, as if asymmetric body plans are just one of the research subjects for proving the importance of new genes. It is fine if they at least compared the roles of new and old genes in the context of asymmetric body plans, but I did not find them in the manuscript.

                 In addition, I found that the analyses and discussion on the promotion of asymmetric body plans are very rough. I don’t find any justification why they used Japanese flounder (left-eyed flatfish) in the transcriptome data analysis whereas starry flounder (both left- and right-eyed) was used in the gene expression analysis. Actually, it would have been much more informatic if they investigated transcriptome data in both left- and right-eyed starry flounder. Thus, it is unclear what roles the new genes the authors argued in this manuscript had in evolution of flatfish body plans.

                 It is also noteworthy that almost all organisms have some degrees of asymmetry in their body plans. I agree that flatfishes are unique in their body plans, but one should pay attention the fact above when discussing the origin of asymmetric body plan in general.

Line 18. “in a cold blood vertebrate taxon” – it came from nowhere. Any justification about that?

Line 163. It is unclear what should be mainly concluded from the gene expression analysis. Is it the number of genes expressed in different tissues?

Line 200. If the authors want to know what happened at the origin of asymmetric body plan, isn’t it reasonable to focus more on the genes in the common ancestor of Pleuronectoidei (branch-2)? At the same time, careful discussion is required because not all Pleuronectoidei species have the same body plan (i.e., they could have both left- and right-eyed bodies in the same species and populations).  

Fig. 1. No description regarding the blue- and green-colored numbers in the figure legend. It should be self-explanatory.

Fig. 3. The figure is not self-explanatory. For example, what does each color represent in Fig. 3A?

Reviewer 2 Report

General comments:

The paper by Li and colleagues tackles the evolutionary genetics of flatfishes – a remarkable fish group that has gained attention in systematics and developmental biology. Flatfishes present one of the most intriguing adaptations among vertebrates—an asymmetric external body plan, with both eyes laying on one side of the head. This unique feature establishes flatfishes as one of the best non-model organisms to advance our understanding of the mechanisms regulating the L-R axis in vertebrates. The authors reanalyzed a previously published dataset comprising eight genomes from which four represent flatfish lineages.

The authors used the synteny-based pipeline to identify new genes originating within Pleuronectoidei via different mechanisms. This analysis provides an idea of the number of Pleuronectodei specific genes. However, the results presented are not enough to determine if those genes are involved in the origin of the unique body plan of flatfishes or the result of adaptive evolution within more specific flatfish lineages (see comment about Fig. 1 below). A more exhaustive characterization of these genes is needed for a deeper understanding of this process. In a nutshell, their results are slightly relevant to understanding the evolution of the asymmetric body plan, but the interpretation of the results is superficial at best and lacks novelty.

Another major problem concerns the monophyly of flatfishes – a hot topic in fish systematics with profound implications for the origin of the asymmetric body plan. Unfortunately, the information presented by the authors is superficial and disregards a substantial body of work that addresses this issue from different perspectives. Indeed, two of the most recent systematic studies addressing flatfish monophyly provide support for the null hypothesis – i.e., that a sister relationship between Psettoidei and Pleuronectoidei exist, suggesting that the asymmetric body plan has a single evolutionary origin. These studies use genome-wide data obtained from a large number of species and provide an extensive examination of morphological characters supporting flatfishes as a natural group. Although the phylogenetic scope of their the current study is restricted to pleuronectoids, failure to consider the possibility that Psettodes could indeed be sister to Pleuronectoidei, has profound implications for our understanding of the evolutionary origin of the asymmetric body plan. The evidence considered by the authors for the non-monophyly of flatfishes is far from compelling.

Specific comments:

Line 47: The use of the term "real-flatfish" implies the existence of a "fake-flatfish" lineage. I recommend modifying the wording in this sentence since this is a contentious topic in fish evolution.

Line 57: The authors shoulf clarify that the current study did not generate new data but instead reanalyzed previously published data.

Line 65: This section lacks essential information regarding the nucleotide matrix's size and completeness (how many loci and tips were used in these analyses?)

Line 72: The authors did not follow the best practices for justifying fossil calibrations in phylogenetic trees (See Parham et al. Syst Biol. [2012]). A list of fossils used and the fossil placement justifications is required.

Line 92: The method used to infer the mechanisms of origination of new genes is poorly described. Instead, the authors cite studies that either developed or used this method. This section is a crucial component of the manuscript that requires more detail. 

Line 114: Why did the authors use different Ks/Ka thresholds to identify paralogous versus de novo new genes (Ks/Ka < 0.5 versus Ks/Ka < 1)?

Line 157: I am not sure how the SBP results for Pleuronectoidei relate to the retention of genes originating during the ancient events of "fish-specific genome duplication." There is at least one example of a gene (cmn-elastin) that has gone through significant modification after the whole-genome duplication in teleosts, and it has a critical function: regulating the spine formation and patterning in fish. How do your results support that "ancient fish-specific genome duplication event may not retain many duplicates in the modern teleosts?"

Line 164: The authors have analyzed transcriptomic data for two different species in this study (Platichthys stellatus and Paralichthys olivaceus). It is unclear whether the authors used both transcriptomic datasets to identify new genes that are expressed or if they limited their analysis to P. stellatus

Figure 1: According to this figure, the distribution of the origination of new genes is temporally unbalanced, with fewer genes originating early in the evolution of Pleuronectoidei (a total of 6 new genes are present in all Pleuronectoidei; branch 2) while 1341 new genes (87% of the total amount of new genes recovered) are exclusive to Platichthys stellatus (branch 5). This pattern suggests two alternative scenarios: 

1) Some of these 1341 genes may be present in other flatfish lineages but were not recovered in the analyses due to genomic sampling biases (the genome of Platichthys stellatus is of greater quality)

2) Most of the new genes are associated with adaptive evolution within the family Pleuronectidae but not necessarily involved in the process of eye migration or loss of the external body symmetry in the flatfish ancestor. 

Line 174: The age reported for the crown Pleuronectoidei is much older than the age previously reported for the same group. Harrington et al. BMC Evolutionary Biology (2016) provides the most comprehensive fossil calibration for the origin of flatfishes, estimating the pleuronectiform crown node (Psettoidei + Pleuronectoidei) at 61.3 Mya (95 % HPD: 54.3–69.5 Mya). Harrington's results are in line with the oldest fossils ever reported for flatfishes (Amphistium and Heteronectes; both from Monte Bolca, Italy, dated at ~50 Mya). 

Line 199: Ka/Ks analysis for de novo new genes are limited to P. stellatus and P. olivaceus. As such, the interpretation of these results should not be extended to all Pleuronectoidei lineages.

Line 249: As briefly mentioned in this paragraph, the role of the Hipk gene family as part of the metabolic pathway responsible for the eye migration in flatfishes has already been discussed by Shao et al. Nature Genetics (2017; the same study that generated some of the genomic and transcriptomic data reanalyzed in the present tudy). Shao and colleagues were the first to detect the Hipk gene family expansion in Pleuronectoidei (21 copies in P. olivaceus and 22 copies in C. semilaevis, compared to 5-8 among Teleost outgroups). That being said, the discussion around the evolution of the Hipk gene family and its relationship with the loss of body symmetry in flatfishes lacks novelty and depth in the current manuscript. Given the importance of this gene family and the slightly broader taxonomic coverage of the present study, I would expect, for example, a discussion tracing the gene duplication events on the phylogeny. Where and when did the duplication events happen? Did it happen just once of multiple times within different Pleuronectoidei lineages?